# Potential impact of intervention strategies on COVID-19 transmission in Malawi: a mathematical modelling study

Tara Mangal ,[1] Charlie Whittaker,[1] Dominic Nkhoma,[2] Wingston Ng'ambi,[2] Oliver Watson ,[1] Patrick Walker,[1] Azra Ghani,[1] Paul Revill,[3] Timothy Colbourn,[4] Andrew Phillips,[5] Timothy Hallett,[1] Joseph Mfutso-Bengo[2]

Extraordinary Think Tank Meeting on COVID-19, University of Malawi Health Economics and Policy Unit (HEPU). May 2020

[1]Infectious Disease Epidemiology, Imperial College London, London, UK
[2]College of Medicine, University of Malawi, Lilongwe, Malawi
[3]Centre for Health Economics, University of York, York, UK
[4]Institute for Global Health, University College London, London, UK
[5]HIV Epidemiology and Biostatistics Group, University College London, London, UK

**Correspondence to**
Dr Tara Mangal;
t.mangal@imperial.ac.uk

## ABSTRACT

**Background** COVID-19 mitigation strategies have been challenging to implement in resource-limited settings due to the potential for widespread disruption to social and economic well-being. Here we predict the clinical severity of COVID-19 in Malawi, quantifying the potential impact of intervention strategies and increases in health system capacity.

**Methods** The infection fatality ratios (IFR) were predicted by adjusting reported IFR for China, accounting for demography, the current prevalence of comorbidities and health system capacity. These estimates were input into an age-structured deterministic model, which simulated the epidemic trajectory with non-pharmaceutical interventions and increases in health system capacity.

**Findings** The predicted population-level IFR in Malawi, adjusted for age and comorbidity prevalence, is lower than that estimated for China (0.26%, 95% uncertainty interval (UI) 0.12%–0.69%, compared with 0.60%, 95% CI 0.4% to 1.3% in China); however, the health system constraints increase the predicted IFR to 0.83%, 95% UI 0.49%–1.39%. The interventions implemented in January 2021 could potentially avert 54 400 deaths (95% UI 26 900–97 300) over the course of the epidemic compared with an unmitigated outbreak. Enhanced shielding of people aged ≥60 years could avert 40 200 further deaths (95% UI 25 300–69 700) and halve intensive care unit admissions at the peak of the outbreak. A novel therapeutic agent which reduces mortality by 0.65 and 0.8 for severe and critical cases, respectively, in combination with increasing hospital capacity, could reduce projected mortality to 2.5 deaths per 1000 population (95% UI 1.9–3.6).

**Conclusion** We find the interventions currently used in Malawi are unlikely to effectively prevent SARS-CoV-2 transmission but will have a significant impact on mortality. Increases in health system capacity and the introduction of novel therapeutics are likely to further reduce the projected numbers of deaths.

## Strengths and limitations of this study

► This is the first study to date which combines country-specific infection fatality ratios (IFRs) of COVID-19 for Malawi adjusted for comorbidity prevalence and with consideration of the prevailing health system constraints and the impact of these constraints on mortality rates.

► An age-structured deterministic model was used to characterise the spread of SARS-CoV-2 throughout Malawi using the Malawi-specific adjusted IFR.

► The impacts of non-pharmaceutical interventions, novel therapeutics and hospital capacity were analysed, and the effects on incidence, mortality and hospital demand are presented.

► The IFR used as the baseline, from which we inferred the adjusted IFR for Malawi, and some key parameters used in the simulation modelling relied on data from outside sub-Saharan Africa due to the limited numbers of cases there and might therefore not be directly transferable to Malawi.

## INTRODUCTION

As of 15 March 2021, the novel coronavirus SARS-CoV-2 had spread throughout every continent, with over 100 million cases and 2.5 million deaths reported worldwide.[1] Case numbers in the African continent continue to rise, and until widespread deployment of an effective vaccine, there is a critical reliance on non-pharmaceutical interventions (NPI) to reduce transmission. These measures include isolation of suspected/confirmed cases, contact tracing, social distancing, travel restrictions, face covering, school and workplace closures, and shielding of the most vulnerable.[2–5]

By 1 January 2021, Malawi was experiencing a second wave of infections, and additional restrictions were placed on the population. Schools and workplaces were closed in many districts; large gatherings and public events were banned; but a full lockdown has been prohibited due to concerns around the implications on vulnerable populations.[6] The impact of NPI on SARS-CoV-2 transmission in Malawi depends critically on the local context such as population behaviour (including uptake of and compliance with such measures), population movement and contact patterns (Malawi is over 80% rural

and many rely on subsistence farming) and health system capacity.[7]

The impact of NPI can be summarised as a change in the effective reproduction number $R_t$, which represents the average number of secondary infections resulting from one infected case. Studies based on high-income countries have shown that strict interventions such as lockdown, where population movement is limited to essential travel and most public facilities and transport links are closed, have shown the most success in reducing transmission.[3 4 8–10] However, major restrictions to working practices or public transport may have catastrophic implications in sub-Saharan Africa, where many have limited financial capacity to withstand income shocks and no access to social protection programmes.[11] Face coverings have been recommended by WHO as one possible intervention which could reduce transmission of SARS-CoV-2 with minimal socioeconomic implications despite a lack of high-quality evidence.[12 13] Nevertheless, many countries (including Canada, South Korea and the UK), have made face coverings mandatory in public spaces.

A key priority during this emerging pandemic is estimating clinical severity and health system requirements. Current oxygen capacity in hospitals may not be sufficient to give supportive care to large numbers of severe COVID-19 cases; 65% of 34 hospital wards in Malawian hospitals recently assessed had (what was considered pre-COVID-19) adequate access to oxygen and priority is now being given to scaling up access and supplies urgently.[14 15] Early studies suggest oxygen supplementation could reduce the need for mechanical ventilation and lower the risk of death.[16] Additionally, several therapeutic agents, such as tocilizumab and dexamethasone, have been effective in improving patient outcomes, although none so far have been widely tested in sub-Saharan Africa.[17 18]

Infection fatality ratios (IFR), defined as the number of deaths divided by the number of infections, are challenging to estimate, particularly in an emerging outbreak, due to the difficulties in identifying the true number of infected people (both asymptomatic and symptomatic). IFR are strongly dependent on age, and the majority of deaths reported early in the epidemic were among those aged over 60 years.[19–21] There is also a growing body of evidence on the elevated risk of mortality with certain underlying comorbidities, such as cardiovascular disease (CVD), diabetes, chronic obstructive pulmonary disease (COPD) and infectious diseases, for example, HIV, tuberculosis (TB) and malaria.[22–25] The majority of these data were reported from high-income settings, which have borne the highest burden of COVID-19 disease recorded so far. It is not yet clear how these risk factors will affect COVID-19 severity in countries like Malawi, which have a younger population overall, but a high prevalence of infectious diseases and untreated chronic conditions.

## Objectives

Three objectives form the focus of this paper: (1) to predict disease severity caused by SARS-CoV-2 in the Malawian population, given its demographic structure, the prevalence of key comorbidities and health system capacity; (2) to examine the potential impact of a range of NPI that have been or could be used in Malawi; and on that basis, (3) to investigate the potential extent to which increasing health system capacity and/or providing therapeutics could contribute to reducing deaths due to SARS-CoV-2 infection.

## METHODS

We present the methods in three sections that relate to each of our aims.

### Predictions of IFR in Malawi

Our approach uses data on age-specific IFR from China (one of the few studies which applies demography-adjusted underascertainment corrections) and then makes adjustments based on the demography and relative burdens of diseases relevant to COVID-19 risk between China and Malawi, making assumptions about the extent to which each disease affects IFR and the extent and impact of healthcare available.[21] First, we predict IFR by age under the assumption of similar availability and impact of healthcare. We then use these predicted IFR and adjust for the potential impact of a constrained healthcare system in Malawi, making assumptions on the effect of treatment on mortality rates of severe and critical cases. The predicted IFR therefore represents pooled estimates of those receiving and not receiving care.

### Predicted IFR with an unconstrained health system

The prevalence of HIV (virally suppressed and unsuppressed), active TB, clinical malaria, CVD, COPD, hypertension, diabetes (types I and II), obesity (defined as Body Mass Index of $\geq 28\,\text{kg/m}^2$ according to Chinese criteria and $\geq 30\,\text{kg/m}^2$ using Malawian criteria) and malnutrition were extracted for Chinese and Malawian populations (see online supplemental figure 1 and online supplemental table 1 for data sources). We created a unified risk factor for 'metabolic syndrome', defined as the presence of at least one of the following conditions which tend to be clustered within individuals: CVD, hypertension, obesity and diabetes. The plausible range for the risks of mortality due to metabolic syndrome was taken as the outer bounds of the relative risks reported for each of the pooled conditions. Given the considerable uncertainty in these estimates along with likely differences across settings, we sample from a wide range of relative risk values for each comorbidity (online supplemental table 2).

The baseline age-distributed IFR were derived from those published by Verity *et al* for cases reported in mainland China, using linear interpolation on the log scale to derive values in 5-year age groups from the 10-year age

bands reported.[21] Adjusted IFR by age for Malawi were computed as follows:

1. Lognormal distributions were derived for each of the age-distributed IFR from China such that the mean matched the mean IFR, and 95% of the probability mass fell inside the reported 95% bounds. Uniform distributions were defined for the relative risks of mortality due to COVID-19 for each comorbidity covering the range described in online supplemental table 2.
2. Age-specific IFR and relative risks of mortality with comorbidities were sampled from the defined distributions.
3. Age-distributed IFR for a theoretical population with no comorbidities ($IFR^*_a$) were computed using the sampled values as follows:

$$IFR^*_a = \frac{IFR_{h=China,a}}{\sum_i (r_i \cdot c_{i,h,a})} \qquad (1)$$

where $IFRh_{h=China,a}$ was the sampled IFR in setting $h$ (where $h$ is China); $i$ is the index for each comorbidity; $r_i$ is the sampled relative risk of mortality for each condition; and $c_{i,h,a}$ is the prevalence of each comorbidity in setting $h$. All terms except relative risk values were indexed by age group $a$, assuming that there are no interactions between age and relative risk of death due to comorbidity.

4. The adjusted IFR for Malawi were then estimated using equation 2:

$$IFR_{h=Malawi,a} = IFR^*_a \sum_i \left( r_i * c_{i,h=Malawi,a} \right) \qquad (2)$$

5. Steps 2–4 were repeated 1000 times and the median adjusted IFR and uncertainty intervals (UIs) were calculated as the 50th, 2.5th and 97.5th quantiles from the sampled estimates.

A summary for the average IFR for Malawi was obtained by weighting the age-specific IFR by the proportion of the population in each age group, as follows:

$$IFR_{h=Malawi} = \frac{\sum_a \left( N_a . IFR_{h=Malawi,a} \right)}{\sum_a N_a} \qquad (3)$$

where $N_a$ is the number of persons in that age group. The analysis was repeated using data on IFR from Brazil to determine whether the choice of primary data would affect the predicted IFR in Malawi.

This method assumed that the differences in IFR between settings are driven by the presence of comorbidities and differences in the age structure of the population.

## Predicted IFR adjusting for health system constraints

The effect of access to and quality of healthcare was accounted for through the following additional steps:

1. Parameter sets defining the proportion of COVID-19 cases requiring different levels of hospital care (severe or critical) were generated using rejection sampling on the basis of (1) prior information from high-income settings and (2) assumptions for mortality according to disease severity with treatment (online supplemental table 3).

2. These estimates of disease severity were used in the simulation model (detailed in the following section) to simulate epidemic trajectories over 365 days without NPI, where the availability of hospital care was limited to the level currently prevailing in Malawi.
3. The induced overall IFR in Malawi, given the current healthcare system constraints at the end of the epidemic, was 'number who died of COVID-19/number ever infected with SARS-CoV-2'.

Further details are provided in the Supplementary Information.

The resulting population-level IFR predictions for Malawi were compared with China using Monte Carlo simulation. IFR values for China were sampled 1000 times from the lognormal distributions for comparison and the Kolmogorov-Smirnov (KS) test statistic was computed, producing a KS test statistic distribution.

## Estimates of the potential impact of NPI on transmission of SARS-CoV-2

The COVID-19 model of Walker *et al* was used to make projections of the spread of SARS-CoV-2 in Malawi under a number of NPI scenarios.[26] Briefly, the deterministic model comprises an age-structured Susceptible-Exposed-Infected-Recovered (SEIR) compartmental framework which describes the transmission of SARS-CoV-2 through an otherwise homogenous population. The contact rates between age groups were derived from the Manicaland study in Zimbabwe (online supplemental table 4).[27] Infected (and infectious) cases were classified as mild (not requiring care), severe (requiring hospitalisation/oxygen) and critical (requiring intensive care unit (ICU)/mechanical ventilation) with the likelihood of receiving care constrained by the prevailing health system capacity. We assumed that not all critical cases required mechanical ventilation, but for those that do, access to an ICU bed also indicates availability of mechanical ventilation. In each infected stage, there was a probability of death, derived from multinational analyses from data in China, the UK and the USA. It was assumed that hospitalised cases would not contribute to transmission. In the case of a person needing care but the health system capacity being exhausted, the person was exposed to a risk of death consistent with no care being received (online supplemental table 5).

### Epidemic setting

We assumed $R_t=2$, which was a central value for the estimates of $R_t$ in Malawi immediately prior to the second wave and varied this between 1.5 and 2.5 to reflect the uncertainty in this assumption.[28] At the start of the simulation, 20 cases were seeded in age groups 35–54, reflecting the ages of those who were most likely to be working or travelling and acquired infection. The population was considered to be fully susceptible during this simulated second wave due to waning immunity following infection in the first wave, along with the emergence of

**Table 1** List of the interventions under consideration along with their implementation in the model

| Strategy | Implementation |
|---|---|
| Current situation | Assumed to be in place at the start of the outbreak<br>Workplaces are closed; public events are banned; restrictions on gatherings are in place; public transport is reduced.<br>Consider these strategies as a bundle equating to a combined reduction in $R_t$ of 24%.[9]<br>Dates and details of individual non-pharmaceutical interventions are reported by OxCGRT.[6] |
| Enhanced shielding:<br>Current situation, plus<br>shielding of those aged ≥60 years | Reduce contact rates by 60% for populations aged ≥60 years in addition to the reduction in $R_t$ mentioned previously.<br>It is implemented after a trigger is reached.* |
| Lockdown:<br>Current situation plus:<br>Stay-at-home requirements<br>School closures<br>Enforcement of social distancing in excepted businesses<br>Prohibition of public transportation<br>Prohibition of all gatherings outside household | Consider that this bundle equates to a sustained reduction in $R_t$ of 42%.[9]<br>It is implemented after a trigger is reached.* |

*The trigger date for interventions to be applied was when the rate of death exceeded 1.0 COVID-19 deaths per 100 000 population per week.

newer strains capable of evading pre-existing immune responses.[29]

### NPI strategies

The NPIs that were in place at the start of the second wave are summarised in table 1. We additionally considered the potential impact of shielding those aged ≥60 years and lockdown. The duration of lockdown varied between 6 and 24 weeks, whereupon the previously implemented intervention strategies were resumed. The trigger day for the implementation of NPI was when the rate of deaths exceeded 1.0 death per 100 000 population per week.

Face coverings were analysed as an incremental intervention on top of the existing measures, and we explored a full range of values for efficacy and proper usage, assuming that the current measures in place remained for the duration of the simulation. We did not distinguish between household and non-household transmission and assumed adherence and efficacy jointly reduced the risk of transmission to the whole population.

The low numbers of deaths reported to date in Malawi coupled with the high potential for under-reporting make formal calibration to surveillance data problematic. We opted, therefore, to present a hypothetical scenario using plausible estimates for transmission rates in this setting and incorporating uncertainty around key assumptions.

We ran each simulation 1000 times over 365 modelled days using the sampled parameter sets for disease severity (see previous discussion). The outcomes of each intervention on the daily number infected, health system requirements (broken down by severity) and deaths are presented.

### Estimates of the impact of increasing health system capacity

The projected impact of increasing the number of non-intensive care hospital beds plus availability of oxygen and the introduction of a novel therapeutic agent were examined, assuming that the current intervention strategies would remain in place indefinitely. We simulated an increase in hospital bed capacity (plus access to oxygen) by up to 100% from the trigger day of when the rate of deaths exceeded 1.0 deaths per 100 000 population per week, presenting the resulting impact on the cumulative number of deaths projected to occur over the epidemic. Additionally, the impact of a novel therapeutic agent was analysed, assuming a proportional reduction in mortality for severe and critical cases (0.65 and 0.8, respectively, which is of the same order of magnitude as indicated in the Randomised Evaluation of COVID-19 Therapy (RECOVERY) trial on dexamethasone, but the modelled agent is hypothetical) applicable to all age groups.[17] We assumed that the therapeutic agent could be administered to those in need even if hospital beds or ICU beds were not available.

All analyses were conducted in R statistical software V.3.6.3 (https://www.r-project.org/).

### Data sharing agreement

Source code and supporting documentation for the SEIR model are available online (https://githubcom/mrc-ide/squire). All data used in the analyses are publicly available and sources have been listed for each.

### Patient and public involvement

Patients and/or the public were not involved in the design, conduct, reporting or dissemination plans of this research.

### Role of the funding source

The funders of this study had no role in study design, data analysis, data interpretation or writing of the report. All

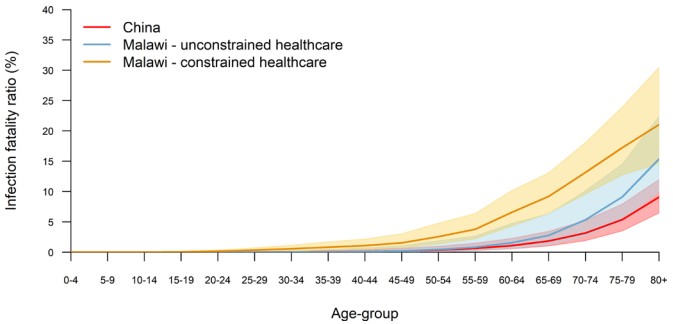

**Figure 1** Predicted infection fatality ratios for the Malawian population with unconstrained and constrained healthcare (according to current health system capacity) compared with estimates reported from China.

authors had access to all data in the study and accept final responsibility for the decision to submit for publication.

## RESULTS
### Predictions of infection fatality ratios in Malawi
Under the assumption of similar healthcare availability, the predicted age-specific IFR for Malawi are higher for every age group than those reported in China. However, the predicted population-weighted IFR is lower (IFR 0.26%, 95% UI 0.12%–0.69%, compared with 0.6%, 95% CI 0.40% to 1.30% in China) due to the younger average age of the population (figure 1). Incorporating

health system constraints through the simulation model results in significantly higher age-specific IFR for Malawi (p<0.05, KS test for all age groups), although the population-weighted estimate is not significantly different from that reported for China (overall IFR 0.83%, 95% UI 0.49%–1.39%). When using the Brazil data, the adjusted population-weighted IFR for Malawi assuming no health system constraints is 0.48% (95% UI 0.33%–0.64%), compared with 0.26% (95% UI 0.12%–0.69%) when using the Chinese data (online supplemental figure 2).

### Estimates of the potential impact of NPI on transmission of SARS-CoV-2
The projected unmitigated scenario is presented as counterfactual, showing what could occur had no interventions been introduced (figure 2). With the current interventions in place and assumed to be in place indefinitely, we estimate approximately 54 400 deaths (95% UI 26 900–97 300) could be averted over the course of the epidemic compared with an unmitigated scenario in which 134 300 deaths (95% UI 82 100–222 500) are projected to occur (table 2). Enhanced shielding of people aged ≥60 years could avert a further 40 200 deaths (95% UI 25 300–69 700) and halve ICU admissions at the peak of the outbreak. These measures would also delay the spread of infection, shifting the peak in infections by approximately 66 days.

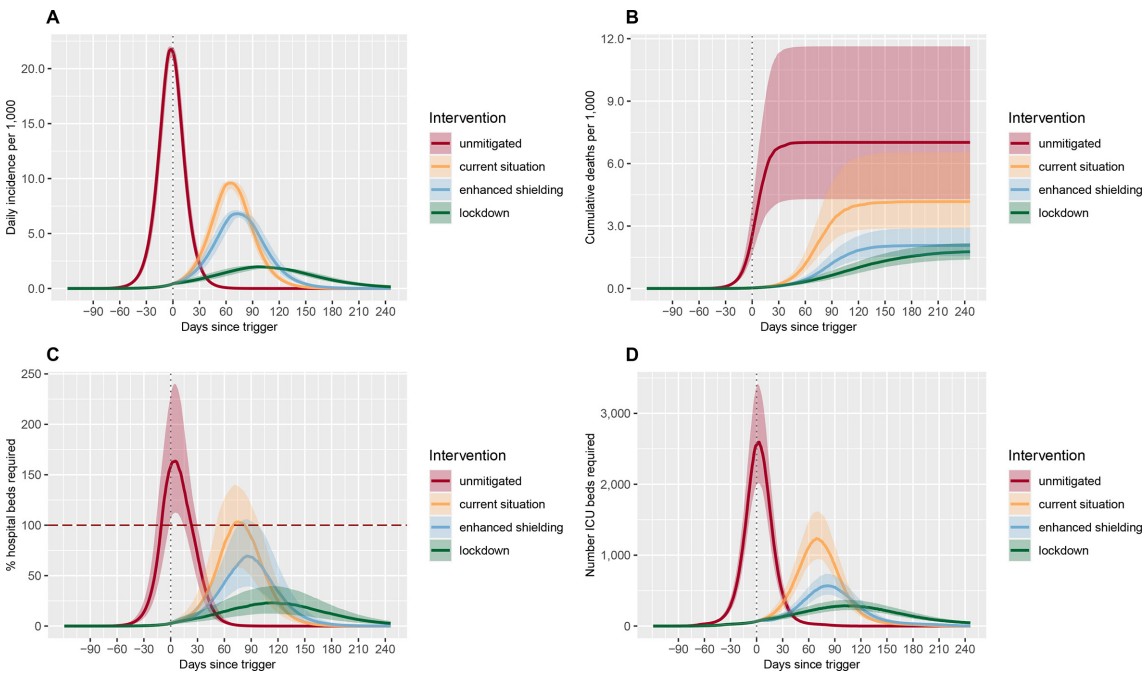

**Figure 2** Impact of NPI compared with a baseline (unmitigated) scenario on the daily incidence per 1000 population (A), the cumulative deaths per 1000 population (B), the percentage of hospital beds that are required (C) and the number of ICU beds that are required (D). The unmitigated scenario represents the counterfactual situation had no interventions been introduced. The current situation reflects the NPI adopted by Malawi at the start of the second wave. Enhanced shielding refers to reducing contact rates of people aged ≥60 years. Lockdown is the adoption of stringent social distancing policies. Further details are presented in table 1. The trigger date is shown with a vertical grey dashed line. The red horizontal dashed line shows the capacity of the health system for non-intensive care (C). ICU capacity comprises 25 ICU beds and 16 mechanical ventilators. ICU, intensive care unit; NPI, non-pharmaceutical intervention.

**Table 2** Outputs from intervention strategies over 365 days

|  | Unmitigated | Current situation | Enhanced shielding | Lockdown |
|---|---|---|---|---|
| Total cases/1000 population | 769.7 (668.4–872.6) | 575.3 (483.1–667.4) | 499.8 (412.4–591.4) | 275.0 (223.6–326.9) |
| Number of general hospital beds required at peak | 40 700 (28 000–59 700) | 25 700 (14 500–34 800) | 17 300 (9800–26 400) | 5700 (3100–9900) |
| Number of ICU beds required at peak | 2600 (2000–3400) | 1200 (900–1600) | 600 (400–700) | 300 (200–400) |
| Total deaths/1000 population | 7.0 (4.3–11.6) | 4.2 (2.9–6.5) | 2.1 (1.6–2.9) | 1.8 (1.4–2.2) |

All values are medians of 1000 simulations using the sampled parameter sets for disease severity. Numbers of hospital and ICU beds are rounded to the nearest 100.

The predicted age distribution of infected people at the peak of the epidemic shows that the majority of infections occur in the younger ages (<20 years) which make up >50% of the population and have high contact rates (online supplemental figure 3), although only 1.3% of deaths occur in that group. There is, however, considerable uncertainty around the prevalence and impact of comorbidities such as HIV and malnutrition in these age groups. The majority of projected deaths occur in those aged over 70 years.

Of the mitigation strategies modelled, long-term lockdown has the largest impact, bringing the infection rate down to 275 infections per 1000 population (95% UI 224–327 per 1000 population) and the mortality rate to 1.8 deaths per 1000 population (95% UI 1.4–2.2 per 1000 population, equivalent to 33 800 deaths, 95% UI 26 600–41 900; table 1). Applying lockdown over 6, 12 and 24 weeks delays the peak incidence, reducing the size of the outbreak and maintaining hospital requirements below capacity, which in turn reduces mortality rates (online supplemental figure 4). The impact of each intervention with $R_t$=1.5 and 2.5 is shown in online supplemental table 6).

With current interventions in place, coverage of face coverings would need to exceed 60% (30% with shielding implemented simultaneously) with a minimum efficacy of 50% in order to reduce the projected $R_t$ to below 1 (figure 3).

### Estimates of the impact of increasing health system capacity

Increasing capacity for non-intensive care (general hospital beds with oxygen availability) by 50% reduces the projected mortality rate to 3.7 deaths per 1000 population (95% UI 2.8–5.3) compared with 4.2 (95% UI 2.9–6.5) under the current scenario (online supplemental table 7). Doubling hospital and oxygen capacity could marginally reduce this further to 3.5 deaths per 1000 population (95% UI 2.8–4.7). Introducing a novel therapeutic agent that is capable of reducing mortality by 0.6 for severe cases and by 0.85 for critical cases, in combination with a 50% scale-up in hospital capacity, reduces projected deaths to 2.5 deaths per 1000 population (95% UI 1.9–3.6, figure 4).

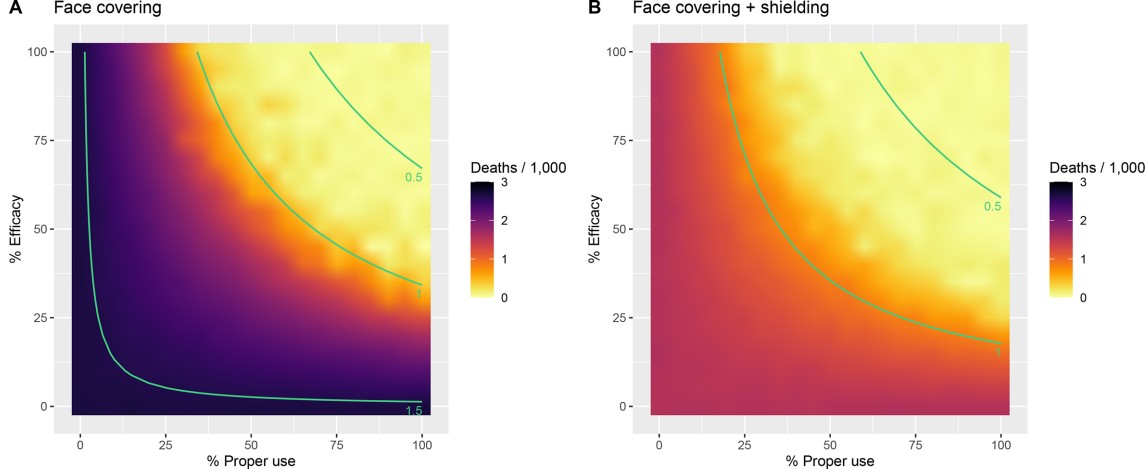

**Figure 3** Impact of face covering (A) and face covering plus enhanced shielding (B) on the total number of deaths per 1000 population projected to occur over 365 days. The full range of values for % efficacy and % proper use (adherence) is presented. The current interventions are assumed to remain in place. The isoclines (green lines) represent the estimated $R_t$, given the efficacy and adherence.

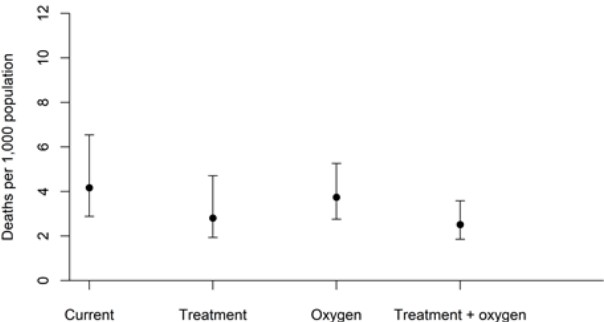

| Intervention | Notes |
|---|---|
| Current | Assumes current interventions remain in place |
| Treatment | Current interventions in place plus availability of a therapeutic agent for all severe / critical cases (irrespective of hospital bed availability) |
| Oxygen | Current interventions in place plus 50% scale-up in hospital beds and oxygen availability |
| Treatment + oxygen | Current interventions in place plus 50% scale-up in hospital beds and oxygen availability and availability of a therapeutic agent for all severe / critical cases (irrespective of hospital bed availability) |

**Figure 4** Projected total numbers of deaths per 1000 population over 365 days with increases in hospital capacity (and oxygen) and a novel therapeutic agent. The points show the median of 1000 simulations, with 2.5th and 97.5th uncertainty intervals represented by the bars.

## DISCUSSION

The results shown here give several important insights into the potential spread and severity of SARS-CoV-2 infection in Malawi and what can be done to prepare for it.

First, under the assumption of similar access to healthcare, overall population-weighted IFR are lower than the reported values for China, consistent with findings from Brazeau *et al* (estimates for low-income countries 0.23, 95% prediction interval 0.14–0.42) due to the younger average age of the population.[21 30] As the availability of healthcare is much lower in Malawi, the overall expectation for IFR increases to 0.83%, 95% UI 0.49%–1.39%, within the range of estimates reported across the Americas, Asia and Europe (overall IFR 0.70, 95% CI 0.57 to 0.82, range 0.28–0.89).[31] Estimates of mortality in Uganda follow a similar trend, with a lower predicted disease burden than the comparator regions (Europe, America and China) and the majority of risk arising indirectly due to disruptions to the health service.[32] Other studies have also reported a differential risk of severe disease in African settings compared with European countries due to underlying health conditions, although there is considerable uncertainty in these analyses.[33 34]

Second, we find that the intervention strategies that have been implemented so far in Malawi would be unlikely to suppress or substantially mitigate the epidemic. Although lockdowns have been highly effective across a number of settings, they may be impractical in Malawi for a number of reasons.[3 35 36] Approximately 50% of the Malawian population lives in poverty, meaning there is no financial buffer if people are unable to earn money.[36 37] The disruptions to food production and delivery chains are likely to impact those who are most vulnerable, with food shortages likely to occur within days of lockdown being implemented.

We find that, similar to other studies, shielding of the older population (plus other vulnerable populations, such as people living with HIV or TB) would be effective in significantly reducing the death rate.[8 38] The practicalities of moving elderly people into separate accommodation in the African context are uncertain at best, but it is a potentially risky strategy as a single imported case

could have devastating effects. In high-income countries, this strategy has proven difficult with outbreaks occurring in many nursing homes.[39] Designating shielded households, or even single rooms within a household may be a viable strategy in Malawi, which has an extensive network of community health workers who could facilitate and support such measures.

Given the difficult choices facing decision-makers in Malawi and elsewhere, it is not surprising that significant attention has turned to less disruptive strategies for restricting some population movement and advocating use of face coverings, a putatively low-cost and readily available means of reducing transmission risk.[40] Delaying the epidemic by even a few months could allow time for new vaccines to be delivered and for further therapeutics to be tested and introduced.[17 41 42] Additional interventions, such as testing (test, trace and isolate) and local quarantining, could have a significant impact on the spread of infection and have been regularly used for case-finding and containment for HIV and TB in low-income and middle-income countries.[43–46] However, they are extremely intensive and difficult to implement at a large scale and therefore may not be a feasible option for mitigating the outbreak in Malawi.

Third, we find that increases in certain types of capacity in the hospital may contribute to reducing deaths. The current priorities for COVID-19 response in Malawi are to distribute vaccines to health workers and significantly expand access to oxygen concentrators.[15 47] There are likely to be improvements to long-term morbidity and lung health if severe cases receive oxygen when required, combined with reduced probability of requiring mechanical ventilation, although we do not capture this here. There is an urgent need for further data to analyse the longer-term impacts of COVID-19 infection to inform this. Incorporating the predicted impact of a vaccine, prioritised for health workers, is not possible using this model as we do not capture occupation-based exposure risks. WHO strongly recommends systemic corticosteroid therapy (dexamethasone) for severe COVID-19 infections, either alone or in combination with other drugs, which may reduce mortality and the need for mechanical ventilation.[41] The impact of these therapies

in sub-Saharan Africa is not yet well documented and is likely to be affected by coinfections, particularly HIV and TB, coupled with late presentation to care.

The potential risk of nosocomial transmission is high, compounded by global shortages in personal protective equipment (PPE).[48] In response, the Ministry of Health in Malawi developed COVID-19 treatment centres away from central hospitals and developed reusable PPE equipment to supplement those already acquired.[49] We optimistically assume here that hospitalised cases are isolated and do not contribute to onwards transmission either in the community or to healthcare workers, which may bias our estimates of disease spread. Additionally, discounting this risk lowers the expected impact of NPI. Other modelling studies have shown variable risks in within-hospital transmission, with Evans *et al* suggesting up to 89% of infections in healthcare workers in England were acquired within the health system and Treibel *et al* finding the majority of these infections were acquired through community transmission.[48 50]

The unadjusted IFR that we use to derive the mortality rates along with some key parameters including treatment outcomes rely on data from high-income and middle-income settings, which may not be directly transferable to Malawi.[3 21 26] Our results are sensitive to the assumptions inherent in these analyses and cannot yet be fully parameterised using data from African settings due to the limited numbers of cases there. Age-structured contact matrices are derived from studies in Zimbabwe, although we expect that there are unlikely to be significant differences between the two settings that would meaningfully affect the conclusions drawn. Household structure is not incorporated; therefore, we cannot adjust for increased risk of infection within households. This may be of particular importance in Malawi, where households are multigenerational and there may not be space to designate separate rooms for high-risk individuals. An important next step in this analysis would be the integration of the model with geographically disaggregated surveillance data on testing, deaths, movement and other data, which could capture local transmission and potentially open the way to more finely targeted interventions that may maximise epidemic control with lesser disruptions overall.

The estimates of relative risk of mortality with comorbidities are derived mainly from studies in high-income or upper-middle-income settings and may vary by age, although we do not capture this here. The management of comorbidities is likely to differ across settings, and so the corresponding risk of mortality with these conditions may vary also. The relative risks of death of the different comorbidities were combined in an additive model, given that the reported HRs used have been adjusted for the presence of other conditions.

The low numbers of cases reported in the first wave of the epidemic in Malawi could be consistent with a lower $R_t$ than is assumed here or an imperfect surveillance system with low numbers of tests being carried out. Approximately 1000 deaths have been reported throughout the whole outbreak, although this is likely an underestimate, with 82% of those occurring in 2021.[47] We estimate here approximately 80 000 deaths may occur if stricter NPI are not introduced, falling to 48 000 if therapeutics effectively moderate mortality rates. Introduction of a vaccine is likely to have a significant impact on the course of the epidemic and, if prioritised to those at highest risk, could substantially reduce the projected number of deaths.

This study has focused on the effects of NPI and health system capacity with respect to one disease, COVID-19. However, imposing lockdown could disrupt routine health services such as the provision of care for HIV, TB and malaria along with national immunisation programmes, compounding increases in mortality rates. Balancing the competing demands on health versus economic productivity, poverty and education is an extremely difficult decision, and we present these projections as a series of hypothetical scenarios which could be used to inform decision-making.

These outputs are not intended to be a forecast of what will happen in Malawi, and evidence should be reviewed in the light of continuously evolving surveillance data and combined with detailed analyses on the broader impacts of any potential interventions. Lessons could be learnt from the South African response, which implemented a rapid, phased strategy, successfully delaying the outbreak despite considerable challenges. In addition to physical distancing and restrictions on movement, South Africa capitalised on existing experienced teams of community health workers to conduct active case finding, along with redirecting contact-tracing teams, previously established for TB control, to conduct COVID-19 contact tracing and monitor quarantine compliance.[44] Clearly, lockdown would have the biggest impact on the spread of SARS-CoV-2, but in settings where this is not feasible, a combination of interventions such as shielding, face covering, increasing hospital capacity and therapeutic agents could together have a significant impact on mortality.

**Contributors** Literature search: TM, TC, AP and TH. Figures: TDM. Study design: TM, CW, OW, PW, ACG, TC, AP, TH and JM-B. Data collection: TM, CW, DN, OW, PW, ACG, TH and JM-B. Data analysis: TM, TH, AP and TC. Data interpretation: TM, CW, DN, OW, PW, ACG, PR, TC, AP, TH and JM-B. Writing: TM, CW, DN, WN, OW, PW, ACG, PR, TC, AP, TH and JM-B.

**Funding** TM, TH, AP, TC, JM-B, PR, DN and WN are supported by UK Research and Innovation as part of the Global Challenges Research Fund (grant number MR/P028004/1). TM, CW, PW, OW, ACG and TH acknowledge joint Centre funding from the UK Medical Research Council and Department for International Development (grant reference: MR/R015600/1). We thank the Imperial College COVID-19 Response Team for their feedback and assistance with method development. We also thank the University of Malawi Health Economics and Policy Unit Think Tank members for their support and collaboration in this project.

**Competing interests** None declared.

**Patient consent for publication** Not required.

**Ethics approval** Ethical approval was not required for this study, which uses publicly available, anonymised data.

**Provenance and peer review** Not commissioned; externally peer reviewed.

**Data availability statement** Data are available in a public, open access repository. Source code and supporting documentation for the SEIR model are available online (https://github.com/mrc-ide/squire). All data used in the analyses are publicly available and sources have been listed for each.

**ORCID iDs**
Tara Mangal http://orcid.org/0000-0001-9222-8632
Oliver Watson http://orcid.org/0000-0003-2374-0741

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
