## [Reviewer comments · BMJ Open]

This paper was submitted to a another journal from BMJ but declined for publication following peer review. The authors addressed the reviewers' comments and submitted the revised paper to BMJ Open. The paper was subsequently accepted for publication at BMJ Open.

(This paper received three reviews from its previous journal but only two reviewers agreed to published their review.)

ARTICLE DETAILS

TITLE (PROVISIONAL)	The potential impact of intervention strategies on COVID-19 transmission in Malawi: A mathematical modelling study
AUTHORS	Mangal, Tara; Whittaker, Charlie; Nkhoma, Dominic; Ng'ambi, Wingston; Watson, Oliver; Walker, Patrick; Ghani, AC; Revill, Paul; Colbourn, Timothy; Phillips, Andrew; Hallett, Timothy; Mfutso-Bengo, Joseph

VERSION 1 – REVIEW

REVIEWER	Khan, Muhammad Ton Duc Thang University
REVIEW RETURNED	05-Oct-2020

GENERAL COMMENTS	The presented some statistical analysis for the COVID-19 transmission in Malawi. The authors mentioned that it is a mathematical study, but i see some less mathematics without a mathematical model. As per this less mathematics which is not including a mathematical model, the results are accepted to be published in this journal. Some important points should need to be addressed.  1. the authors should compare this statical results with an SIR or SEIR type model. 2. In literature there are many mathematical models published on COVID related to the data of each country, it should be compared with this study. some work is under The dynamics of COVID-19 with quarantined and isolation Advances in Difference Equations 2020 (1), 1-22 Modeling the impact of non-pharmaceutical interventions on the dynamics of novel coronavirus with optimal control analysis with a case study Chaos, Solitons & Fractals 139, 110075 Modeling the dynamics of novel coronavirus (2019-nCov) with fractional derivative Alexandria Engineering Journal
--

	3. The graphical results are obtained which shows the method is appropriate and the results are sound.
--	--

REVIEWER	Obolski, Uri Tel Aviv University
-----------------	-------------------------------------

REVIEW RETURNED	09-Jan-2021
-------------

GENERAL COMMENTS	In the study by Mangal et al., the authors project disease burden in Malawai using infection fatality estimates from China, adjusted to the Malawian population. The disease dynamics are simulated using a previously developed deterministic model. They estimate the impact of deployment of non-pharmaceutical public health interventions as well increasing the health system capacity in Malawi on mortality from COVID-19. I think this is a well-written paper with interesting insights, which may be interesting to the readers of BMJ Open. However, I do have some questions and comments that I believe the authors have to address in order for the manuscript to be appropriate for publication. Major comments: • I am not sure I understand the rationale of the metabolic syndrome risk factor definition. First, why would the risk for metabolic syndrome be the highest of the underlying conditions defining it? Since they are necessary conditions, it seems to me that an upper bound on the prevalence of the syndrome would be the minimum prevalence of the underlying conditions defining it; not the maximum, as was used here. Second, the risk for the syndrome was defined as the highest of the risk of the underlying conditions. This could make theoretical sense if you indeed capture the syndrome's prevalence. However, if you don't capture it, then you are essentially taking four risk factors, and assigning the most prevalent risk the highest risk estimate. Seems like a recipe for risk overestimation. Please explain the rationale for doing so.
---

- Why do you use a uniform distribution to sample from the relative risk estimates you present in supplementary table 2? The log relative risk is approximately normally distributed with known variance expressions and it seems this would be more appropriate.
- . “All terms except relative risk values are additionally indexed by age group a” – if I understand correctly, you assume no interactions between age and comorbidities. This should be clearly stated as an assumption (or clearly stated otherwise if I misinterpret) and if possible justified or discussed as a limitation.
- The rejection sampling of the parameters should be more clearly explained. For example, “which fell within the IFR computed for...” - what exactly does falling within the IFR mean?
- "The probability of death in severe cases not receiving treatment is lowered from the default values in the Walker et al model to double that of treated cases to reflect the low mortality observed in Malawi to date" – I am not sure I understand why this is more reasonable than using a different constant than a 2-fold increase in mortality? Also, shouldn't the low mortality in Malawi be reflected in other parameters (e.g. comorbidity/age distribution) rather than being corrected through this specific parameter?
- You state that the IFR in Malawi is lower than that of China, but it is not significantly lower when considering the health care system constraints. Do you mean statistically significant? Did you perform inference on the difference of estimates in both cases? It should be possible through a Monte Carlo simulation of the difference given the distributions of both estimators are known.
- Perhaps I missed it, but I could only find how NPIs decrease interactions. This is not true for certain NPIs. That is, closing schools and workplaces means that people spend more time at home with their family, so certain age-dependent contacts (e.g. of young adults and

	young children) are expected to increase. I believe that there exist some post-lockdown contact matrix estimations in recent article that can help you model these phenomena. Minor comments:  • Lines 108-109 "...mortality is not well established however early studies..." • Line 286 "...by 0.6 for severe cases by 0.85..."
--	---

REVIEWER	Thompson, KM Kid Risk, Inc.
REVIEW RETURNED	10-Jan-2021

GENERAL COMMENTS	1. I commend the authors on a well-written and interesting analysis that provides insights about the potential benefits of the application of non pharmaceutical interventions (NPIs) in reducing the burden of COVID-19 in Malawi. This analysis provides a useful example of the type of national analysis that could be performed for other countries and helpful context for people in Malawi and in other countries with similar conditions. 2. My one substantive comment relates to the sentence on line 196 of the submission related to hospital cases not contributing to transmission. Multiple modeling studies suggest that nosocomial transmission represents an important contribution to transmission for viruses like SARS. I assume that the authors mean to say that they are implicitly assuming: (1) that once hospitalized, patients are isolated and therefore can no longer participate in community transmission and they are distinguishing this group from other infected individuals. I believe that they are implicitly assuming that hospital staff are fully trained on infection prevention, given PPE, and thus are not at increased risk of becoming infected or infecting others in the general population. The assumptions used by the authors to model the experience of the general population are fine, but the authors may wish to at least acknowledge the existence of studies that explicitly consider the role of nosocomial infection and the importance of the health system in isolating cases to prevent community transmission. This may also lead the authors to cite some additional prior modeling papers that demonstrated the impacts of NPIs for other settings that would further help put this work in context. 2. In line 332, I suggest replacing the word "individuals" with the word "infections" so that it reads: "We define infections as mild, ..."
---

REVIEWER	Champagne, Clara Swiss Tropical and Public Health Institute
REVIEW RETURNED	25-Jan-2021

GENERAL COMMENTS

In this manuscript, Mangal et al. provide calculations of the age-specific infection fatality ratios (IFR) for Covid-19 in Malawi and use them within a pre-existing compartmental model to simulate the effect of several non-pharmaceutical interventions and hospital capacity scenarios. The IFR are found to be slightly lower but similar to China's, as the younger population compensates for higher comorbidities prevalence and lower hospital capacities. Several NPI and health care scenarios are simulated and their predicted effects on Covid-19 infections, hospital requirements and deaths in Malawi are reported.

I have however some major comments regarding the presentation of the results and some methodological aspects, as well as minor comments as follows.

Major comments:

1. The authors indicate throughout the manuscript that they "estimate" the IFR for Malawi. In my opinion, the term "estimation" is too strong given the fact that the present methodology does not rely on specific Covid-19 data from Malawi (if I understand correctly, the Malawi-specific data used in this analysis are only the age-specific population sizes, the comorbidity prevalences and the availability of hospital care). For example, the terms "extrapolation" or "prediction" would be more appropriate, especially if the targeted audience of the manuscript goes beyond the community of modellers.

2. Additional details on the IFR calculations would be welcome.

-It is not clear what the index i represents: I did not understand if all comorbidities classified as metabolic syndrome count as one comorbidity or as several ones

-The choice of an additive model for the relative risks of the different comorbidities could be stated more clearly and motivated. The curve of the comorbidities by age $\sum r_i c_{ia}$ for China and Malawi could be shown in the supplement.

-Indicate more precisely why it was decided to use the baseline IFR from China. How do these estimates vary from other estimates in other countries? Could the differences be explained by the differences in comorbidity prevalences?

How sensitive are the results if another country for which data is available was used instead?

-It should be clearly indicated throughout the manuscript in which cases the IFR refers to cases receiving care and in which cases it refers to the average of cases receiving and not-receiving care (also when speaking about the Chinese IFR)

3. The introduction should be updated with more recent information on Malawi's Covid-19 situation and the mitigation measures in place (the information mentioned only goes until the summer of 2020). In particular: has what is called the baseline scenario been maintained?

Several lines (215-216,341, 358-359) in the paper said that the reported deaths in Malawi were low at the time of the analysis. It would be useful to further comment this difference with the predictions from the mathematical model in the discussion.

4. The message from Figure 3 regarding the effectiveness of face covering sounds contradictory with the statements in the introduction and the discussion on the large uncertainties regarding the effects of

face covering. The two should be better articulated.

5. It is very appreciated that the simulation model is publicly available. Additionally, if a specific repository for reproducing the scenarios mentioned in the text as well as prevalence values needed for the IFR calculation was available online, it would greatly enhance the outreach of the work, and enable easier application to other countries.

Minor comments:

1. The description of the 4 NPI scenarios should be included in the methods section of the main text (it is only in Supplementary table 4, and briefly in figure 2's legend).

2. The equation for IRFh in the supplement: why is it multiplied by Na and not the proportion of the population in the a group?

2. The conclusion stated in the abstract ("The risks due to COVID-19 vary across settings and are influenced by age, underlying health and health system capacity.") should be reformulated, as it reflects more the working hypothesis of the analysis (as stated e.g. 1168-169) rather than its result.

3. The sources for the comorbidity prevalence should include the date of the data for all indicators and the average prevalence in the whole population could also be indicated.

4. Similarly, on this fast-evolving topic, it should be ensured that the methods and discussion mention the most recent information/recommendations on the therapeutical agents for severe disease.

5. Legends for Figures 3 and 4 could be more detailed. Especially for fig 4, it could be mentioned what the 2 indicators per category are, and what the uncertainty represents.

6. The term "ICL Global" is mentioned in the supplement but needs to be defined.

7. For some references to reports mainly in the supplement, a link url and an precise title should be included. Similarly, the access dates for websites should be indicated.

VERSION 1 – AUTHOR RESPONSE

Reviewer: 1

Dr. Muhammad Khan, Ton Duc Thang University

Comments to the Author:

The presented some statistical analysis for the COVID-19 transmission in Malawi. The authors mentioned that it is a mathematical study, but i see some less mathematics without a mathematical model. As per this less mathematics which is not including a mathematical model, the results are accepted to be published in this journal. Some important points should need to be addressed.

1. the authors should compare this statical results with an SIR or SEIR type model.

The model is not a statistical model, we do use an SEIR framework for the simulation. We acknowledge that this wasn't clear in the text and have added this to the methods.

Line 186: "Briefly, the deterministic model comprises an age-structured SEIR compartmental deterministic framework..."

2. In literature there are many mathematical models published on COVID related to the data of each country, it should be compared with this study. some work is under

The dynamics of COVID-19 with quarantined and isolation Advances in Difference Equations 2020 (1), 1-22

Modeling the impact of non-pharmaceutical interventions on the dynamics of novel coronavirus with optimal control analysis with a case study Chaos, Solitons & Fractals 139, 110075

Modeling the dynamics of novel coronavirus (2019-nCov) with fractional derivative Alexandria Engineering Journal

We thank the reviewer for highlighting their recent papers on COVID-19 modelling which we had not encountered before. We have now cited the study on NPI along with a number of other newer modelling studies that have been published since the original submission.

3. The graphical results are obtained which shows the method is appropriate and the results are sound.

Thank you

Reviewer: 2

Dr. Uri Obolski, Tel Aviv University

In the study by Mangal et al., the authors project disease burden in Malawi using infection fatality estimates from China, adjusted to the Malawian population. The disease dynamics are simulated using a previously developed deterministic model. They estimate the impact of deployment of non-pharmaceutical public health interventions as well increasing the health system capacity in Malawi on mortality from COVID-19.

I think this is a well-written paper with interesting insights, which may be interesting to the readers of *BMJ Open*.

However, I do have some questions and comments that I believe the authors have to address in order for the manuscript to be appropriate for publication.

Major comments:

- I am not sure I understand the rationale of the metabolic syndrome risk factor definition.

First, why would the risk for metabolic syndrome be the highest of the underlying conditions defining it? Since they are necessary conditions, it seems to me that an upper bound on the prevalence of the syndrome would be the minimum prevalence of the underlying conditions defining it; not the maximum, as was used here.

We have defined "metabolic syndrome" as the presence of at least one of the conditions CVD, hypertension, diabetes and obesity. We agree with the reviewer here, it would be more prudent to use the lowest of the prevalences reported for each age-group and we have changed this in the analysis and made the text clearer in the main manuscript.

Line 136: "We created a unified risk factor for "metabolic syndrome", defined as the presence of at least one of the following conditions which tend to be clustered within individuals: CVD, hypertension, obesity and diabetes. The plausible range for the risks of mortality due to metabolic syndrome were taken as the outer bounds of the relative risks reported for each of the pooled conditions. Given the considerable

uncertainty in these estimates along with likely differences across settings, we sample from a wide range of relative risk values for each comorbidity (Supplementary Table 2)."

Second, the risk for the syndrome was defined as the highest of the risk of the underlying conditions. This could make theoretical sense if you indeed capture the syndrome's prevalence. However, if you don't capture it, then you are essentially taking four risk factors, and assigning the most prevalent risk the highest risk estimate. Seems like a recipe for risk overestimation. Please explain the rationale for doing so.

We thank the reviewer for this feedback and have redefined the risks due to metabolic syndrome. We now use the minimum prevalence across the four conditions included in metabolic syndrome as a conservative estimate of risk. We then use a wide range of relative risk values to characterise the additional risk of mortality due to any of these underlying conditions. This range spans the full range of relative risk values reported for each of the conditions in order to include the widest reasonable uncertainty bounds.

- Why do you use a uniform distribution to sample from the relative risk estimates you present in supplementary table 2? The log relative risk is approximately normally distributed with known variance expressions and it seems this would be more appropriate.

Yes definitely we considered sampling from a parametric distribution for the relative risks. However in some cases, we have combined the uncertainty bounds across multiple studies from multiple settings. Given that we don't have data on relative risks of mortality due to COVID with comorbidities for Malawi, we felt it was appropriate to incorporate the widest reasonable bounds. These data have also been updated to include more recent studies and systematic reviews.

- "All terms except relative risk values are additionally indexed by age group a" – if I understand correctly, you assume no interactions between age and comorbidities. This should be clearly stated as an assumption (or clearly stated otherwise if I misinterpret) and if possible justified or discussed as a limitation.

Yes this is an assumption made in this analysis which we mention in the discussion (line 352). There are as yet no data to inform age-adjusted relative risks of death due to covid-19 by comorbidity.

We now additionally include the following in the methods section to clarify this, line 156:

"All terms except relative risk values are additionally indexed by age-group a under the assumption that there are no interactions between age and relative risk of death due to comorbidity."

- The rejection sampling of the parameters should be more clearly explained. For example, "which fell within the IFR computed for..." - what exactly does falling within the IFR mean? We have worked on the structure and wording of this section in the manuscript to make it clearer and reworded the text in the SI. We appreciate that this was a little confusing and hope that the changes have now improved this.

The phrase "falling within the IFR..." referred to the IFRs arising from the sampled parameter sets for disease severity should fall within the 95% uncertainty interval of the adjusted IFR predicted in the methods section 1 (i).

- "The probability of death in severe cases not receiving treatment is lowered from the default values in the Walker et al model to double that of treated cases to reflect the low mortality observed in Malawi to date" – I am not sure I understand why this is more reasonable than using a different constant than a 2-fold increase in mortality? Also, shouldn't the low mortality in Malawi be reflected in other parameters (e.g. comorbidity/age distribution) rather than being corrected through this specific parameter?

This is a good point and we have changed the mortality rates in severe cases not receiving treatment in the simulation model to the default values from the ICL Global model. This has also been updated in the main text and the SI.

- You state that the IFR in Malawi is lower than that of China, but it is not significantly lower when considering the health care system constraints. Do you mean statistically significant? Did you perform inference on the difference of estimates in both cases? It should be possible

through a Monte Carlo simulation of the difference given the distributions of both estimators are known.

Thank you – this is a really useful point. We have now incorporated MC simulation and KS test to determine whether the two distributions are significantly different. The following text has been added to the methods, line 180:

“The resulting population-level IFR predictions for Malawi were compared with China using Monte Carlo simulation. IFR values for China were sampled 1000 times from the reported distribution and the Kolmogorov-Smirnov (KS) test statistic was computed, producing a KS test statistic distribution.”

- Perhaps I missed it, but I could only find how NPIs decrease interactions. This is not true for certain NPIs. That is, closing schools and workplaces means that people spend more time at home with their family, so certain age-dependent contacts (e.g. of young adults and young children) are expected to increase. I believe that there exist some post-lockdown contact matrix estimations in recent article that can help you model these phenomena.

Many thanks for this point. We have revised the NPI methods now using a recently published article (Li et al, Lancet ID 2021) which estimates the impacts of certain bundles of interventions on Rt. So rather than adjusting Rt and contact matrices separately, with the caveat that we didn't know how they would interact with each other (the impact was assumed to be additive), we now only adjust Rt based on data obtained from 131 countries. The only contact matrix we alter, is to model the effect of enhanced shielding, which is a blanket reduction of 60% for contact rates between >60 year olds and every other age-group.

Minor comments:

- Lines 108-109 “...mortality is not well established however early studies...”

Thank you – corrected now

- Line 286 “...by 0.6 for severe cases by 0.85...”

Thank you – corrected now

Reviewer: 3

Dr. KM Thompson, Kid Risk, Inc.

Comments to the Author:

I commend the authors on a well-written and interesting analysis that provides insights about the potential benefits of the application of non pharmaceutical interventions (NPIs) in reducing the burden of COVID-19 in Malawi. This analysis provides a useful example of the type of national analysis that could be performed for other countries and helpful context for people in Malawi and in other countries with similar conditions.

My one substantive comment relates to the sentence on line 196 of the submission related to hospital cases not contributing to transmission. Multiple modeling studies suggest that nosocomial transmission represents an important contribution to transmission for viruses like SARS. I assume that the authors mean to say that they are implicitly assuming: (1) that once hospitalized, patients are isolated and therefore can no longer participate in community transmission and they are distinguishing this group from other infected individuals. I believe that they are implicitly assuming that hospital staff are fully trained on infection prevention, given PPE, and thus are not at increased risk of becoming infected or infecting others in the general population. The assumptions used by the authors to model the experience of the general population are fine, but the authors may wish to at least acknowledge the existence of studies that explicitly consider the role of nosocomial infection and the importance of the health system in isolating cases to prevent community transmission.

Thank you for raising this important point. We have now added a paragraph in the discussion relating to this point:

“The potential risk of nosocomial transmission is high, compounded by global shortages in personal protective equipment (PPE). In response, the Ministry of Health in Malawi developed COVID-19 treatment

centres away from central hospitals and developed reusable PPE equipment services to supplement those already acquired. We optimistically assume here that hospitalised cases are isolated and do not contribute to onwards transmission either in the community or to healthcare workers, which may bias our estimates of disease spread downwards. Additionally, discounting this risk lowers the expected impact of NPIs. Other modelling studies have shown variable risks in within-hospital transmission, with Evans et al suggesting up to 89% of infections in healthcare workers in England were acquired within the health system and Treibel et al finding the majority of these infections were acquired through community transmission."

This may also lead the authors to cite some additional prior modeling papers that demonstrated the impacts of NPIs for other settings that would further help put this work in context.
We have added several newer modelling citations to the introduction and discussion, in relation to lockdowns, shielding of the elderly and the impact of face masks. We have also updated all situation reports and database extractions.

In line 332, I suggest replacing the word "individuals" with the word "infections" so that it reads: "We define infections as mild, ..."
Thank you – this has been changed

Reviewer: 4

Dr. Clara Champagne, Swiss Tropical and Public Health Institute

Comments to the Author:

In this manuscript, Mangal et al. provide calculations of the age-specific infection fatality ratios (IFR) for Covid-19 in Malawi and use them within a pre-existing compartmental model to simulate the effect of several non-pharmaceutical interventions and hospital capacity scenarios. The IFR are found to be slightly lower but similar to China's, as the younger population compensates for higher comorbidities prevalence and lower hospital capacities. Several NPI and health care scenarios are simulated and their predicted effects on Covid-19 infections, hospital requirements and deaths in Malawi are reported. I have however some major comments regarding the presentation of the results and some methodological aspects, as well as minor comments as follows.

Major comments:

1. The authors indicate throughout the manuscript that they "estimate" the IFR for Malawi. In my opinion, the term "estimation" is too strong given the fact that the present methodology does not rely on specific Covid-19 data from Malawi (if I understand correctly, the Malawi-specific data used in this analysis are only the age-specific population sizes, the comorbidity prevalences and the availability of hospital care). For example, the terms "extrapolation" or "prediction" would be more appropriate, especially if the targeted audience of the manuscript goes beyond the community of modellers.

We have changed this throughout the text to "predicted" rather than "estimated".

2. Additional details on the IFR calculations would be welcome.

-It is not clear what the index i represents: I did not understand if all comorbidities classified as metabolic syndrome count as one comorbidity or as several ones

We have adjusted the text describing the comorbidities and the definition of metabolic syndrome both in the main paper and the supplementary information to make this clearer.

In the methods, line 136: We created a unified risk factor for "metabolic syndrome", defined as the presence of at least one of the following conditions which tend to be clustered within individuals: CVD, hypertension, obesity and diabetes.

Methods, line 154: Where $IFR_{i=China_a}$ was the sampled IFR in setting h (where h is China), i is the index for each comorbidity, r_i is the sampled relative risk of mortality for each condition and $c_{i,h,a}$ is the prevalence of each comorbidity in setting h .

And in Supplementary Table 2 we edited the table to show the conditions which contribute to metabolic syndrome more clearly.

-The choice of an additive model for the relative risks of the different comorbidities could be stated more clearly and motivated. The curve of the comorbidities by age $\sum r_{i c_{ia}}$ for China and Malawi could be shown in the supplement.

The relative risks of death with each disease were taken from studies which adjusted for the presence of other comorbidities. For example, the relative risk of HIV from a study in the Western Cape adjusts for TB, diabetes, hypertension, chronic kidney disease and COPD/asthma. We therefore felt it was appropriate to use an additive model for each of these risks. We have added this to the text, discussion line 355: "The relative risks of death of the different comorbidities were combined in an additive model, given that the reported hazard ratios used have been adjusted for the presence of other conditions."

We have also added a figure in the SI showing the reported prevalence of each condition by age in Malawi and China for reference.

-Indicate more precisely why it was decided to use the baseline IFR from China. How do these estimates vary from other estimates in other countries? Could the differences be explained by the differences in comorbidity prevalences? How sensitive are the results if another country for which data is available was used instead?

We used the baseline IFR from China as, at the time, they were the only robust age-stratified estimates available. More recently, more data have emerged on IFR in other settings, but the only other age-stratified data we have found that does not come from a high-income country is from Brazil. We have therefore repeated the analysis using the estimates from Brazil to see whether this underlying assumption affects our predicted values for Malawi. This is now included in the SI and mentioned in the main text.

-It should be clearly indicated throughout the manuscript in which cases the IFR refers to cases receiving care and in which cases it refers to the average of cases receiving and not-receiving care (also when speaking about the Chinese IFR)

In all cases, the IFR refer to averages of cases receiving and not-receiving care with differing assumptions on the level and outcomes of care available. We realise that this should be made clearer and so have added the following text in Methods, section 1:

"Our approach utilises data on age-specific IFR from China (one of the few studies which applies demography-adjusted under-ascertainment corrections) and then makes adjustments based on the demography and relative burdens of diseases relevant to COVID-19 risk between China and Malawi, making assumptions about the extent to which each disease affects IFR and the extent and impact of healthcare available. First, we predict IFR by age under the assumption of similar availability and impact of healthcare. We then use these predicted IFR and adjust for the potential impact of a constrained healthcare system in Malawi, making assumptions on the effect of treatment on mortality rates of severe and critical cases. The predicted IFR therefore represent pooled estimates of those receiving and not receiving care."

3. The introduction should be updated with more recent information on Malawi's Covid-19 situation and the mitigation measures in place (the information mentioned only goes until the summer of 2020). In particular: has what is called the baseline scenario been maintained?

This has now been updated and the full analysis has been repeated using the current interventions which are in place.

Several lines (215-216,341, 358-359) in the paper said that the reported deaths in Malawi were low at the time of the analysis. It would be useful to further comment this difference with the predictions from the mathematical model in the discussion.

We have now included the following text in the discussion to address this point:

"The low numbers of cases reported in the first wave of the epidemic in Malawi could be consistent with a lower R_0 than is assumed here or an imperfect surveillance system with low numbers of tests being carried out. Approximately 1,000 deaths have been reported throughout the whole outbreak, with 82% of those occurring in 2021. We estimate here approximately 80,000 deaths could be expected if stricter NPI

are not introduced, falling to 48,000 if therapeutics effectively moderate mortality rates. Introduction of a vaccine is likely to have a significant impact on the course of the epidemic, and if prioritised to those at highest risk, could substantially mitigate the predicted death toll.”

4. The message from Figure 3 regarding the effectiveness of face covering sounds contradictory with the statements in the introduction and the discussion on the large uncertainties regarding the effects of face covering. The two should be better articulated.

Figure 3 shows the full range of uncertainty around the effectiveness and adherence of face coverings and we feel that we do convey the uncertainty and assumptions in the messaging from this part of the analysis. It is presented as a sensitivity analysis:

“With current interventions in place, coverage of face coverings would need to exceed 60% (30% with shielding implemented simultaneously) with a minimum of 50% efficacy in order to reduce projected R_t to below 1 (Figure 3).”

5. It is very appreciated that the simulation model is publicly available. Additionally, if a specific repository for reproducing the scenarios mentioned in the text as well as prevalence values needed for the IFR calculation was available online, it would greatly enhance the outreach of the work, and enable easier application to other countries.

Many thanks for this suggestion. We agree that it would be great if other countries could adopt this approach and use our code. We will create a public repository for the source code and documentation.

Minor comments:

1. The description of the 4 NPI scenarios should be included in the methods section of the main text (it is only in Supplementary table 4, and briefly in figure 2's legend).

This table has been moved to the main text.

2. The equation for IRFh in the supplement: why is it multiplied by N_a and not the proportion of the population in the a group?

Thank you for spotting this – it is a typo. It has been corrected now.

2. The conclusion stated in the abstract (“The risks due to COVID-19 vary across settings and are influenced by age, underlying health and health system capacity.”) should be reformulated, as it reflects more the working hypothesis of the analysis (as stated e.g. 1168-169) rather than its result.

We have changed this to the following:

“We find here that the interventions currently used in Malawi are unlikely to effectively prevent transmission of SARS-CoV-2 but increases in health system capacity and the introduction of novel therapeutics are likely to have a significant impact on mortality rates.”

3. The sources for the comorbidity prevalence should include the date of the data for all indicators and the average prevalence in the whole population could also be indicated.

We have added the dates of the prevalence data to Supplementary Table 1. We didn't feel average prevalence values were necessary to add here as they are not used in the analysis. We have added a figure which shows the age-dependent prevalence values for Malawi and China in the supplementary information (supplementary figure 2).

4. Similarly, on this fast-evolving topic, it should be ensured that the methods and discussion mention the most recent information/recommendations on the therapeutical agents for severe disease.

Many thanks for this point. We have updated each section of the manuscript with the latest data on not only the therapeutic agents, but also the estimated impact of NPI, the interventions in place in Malawi, more recent IFR data and up-to-date situation analysis reports.

5. Legends for Figures 3 and 4 could be more detailed. Especially for fig 4, it could be mentioned what the 2 indicators per category are, and what the uncertainty represents.

The legend for figure 3 has now been updated:

“Figure 3. Impact of face covering (A) and face covering plus enhanced shielding (B) on the total number of deaths per 1,000 population projected to occur over 365 days. The full range of values for % efficacy and % proper use (adherence) are presented. The current interventions are assumed to remain in place. The isoclines (green lines) represent the estimated R_t given the percentage efficacy and percentage of proper use (adherence).”

Figure 4 has been updated following a suggestion from another reviewer and now has only one set of results (we removed the different assumptions around mortality in severe cases). We have also updated the legend:

“Figure 4. Projected total numbers of deaths per 1,000 people over 365 days with increases in hospital capacity (and oxygen) and a novel therapeutic agent. The points show the median of 1000 simulations, with 2.5th and 97.5th uncertainty intervals represented by the bars.”

6. The term “ICL Global” is mentioned in the supplement but needs to be defined.

We have changed this now to cite the COVID-19 Global Model with a reference.

7. For some references to reports mainly in the supplement, a link url and an precise title should be included. Similarly, the access dates for websites should be indicated.

Many thanks – this has been updated now.

VERSION 2 – REVIEW

REVIEWER	Khan, Muhammad Ton Duc Thang University
REVIEW RETURNED	27-Mar-2021
GENERAL COMMENTS	The title suggests the mathematical study, but the authors do not address well my comments and hence no comparison and so it should not be fit for publications.
REVIEWER	Obolski, Uri Tel Aviv University
REVIEW RETURNED	29-Mar-2021
GENERAL COMMENTS	The authors have answered all my comments thoughtfully and implemented the necessary corrections. I am more than happy to recommend this paper for publication.
REVIEWER	Champagne, Clara Swiss Tropical and Public Health Institute
REVIEW RETURNED	31-Mar-2021
GENERAL COMMENTS	Many thanks to the authors for their detailed response and the updated manuscript: all my comments have been addressed